# Rapid planning and analysis of high-throughput experiment arrays for reaction discovery

Babak Mahjour [1], Rui Zhang[2], Yuning Shen[1], Andrew McGrath [1], Ruheng Zhao[1], Osama G. Mohamed [3], Yingfu Lin [1], Zirong Zhang[1], James L. Douthwaite [1], Ashootosh Tripathi [1,3] & Tim Cernak[1,2] ✉

High-throughput experimentation (HTE) is an increasingly important tool in reaction discovery. While the hardware for running HTE in the chemical laboratory has evolved significantly in recent years, there remains a need for software solutions to navigate data-rich experiments. Here we have developed phactor™, a software that facilitates the performance and analysis of HTE in a chemical laboratory. phactor™ allows experimentalists to rapidly design arrays of chemical reactions or direct-to-biology experiments in 24, 96, 384, or 1,536 wellplates. Users can access online reagent data, such as a chemical inventory, to virtually populate wells with experiments and produce instructions to perform the reaction array manually, or with the assistance of a liquid handling robot. After completion of the reaction array, analytical results can be uploaded for facile evaluation, and to guide the next series of experiments. All chemical data, metadata, and results are stored in machine-readable formats that are readily translatable to various software. We also demonstrate the use of phactor™ in the discovery of several chemistries, including the identification of a low micromolar inhibitor of the SARS-CoV-2 main protease. Furthermore, phactor™ has been made available for free academic use in 24- and 96-well formats via an online interface.

Miniaturized high-throughput experimentation (HTE) has emerged as an accessible, reliable, economical, and environmentally friendly technique for the rapid discovery of new reactivities[1–33]. Curated HTE data has proven to be increasingly valuable for predictive models[15–20]. While in experimental practice, the community has gravitated towards liquid handling techniques in glass shell microvials with tumble stir dowels, or in plastic 384 or 1536 wellplates[1,4–14], a standard for HTE data handling has yet to be established. The organizational load required to perform a simple 24-well reaction array is generally manageable by repetitive notebook entries or with spreadsheets, yet managing multiple reaction arrays in a single day, or running ultraHTE in 1536 wellplates[1], is challenging without information management software.

Moreover, no readily available electronic lab notebook (ELN) can store HTE details in a tractable manner[21,22] or provide a simple interface to extract data and results from multiple experiments simultaneously[23,24]. Contemporary HTE software provides HTE solutions but is only commercially available[34,35]. To continue developing HTE research and position data outputs for machine learning studies, detailed reaction data must be easily accessible for standardized rapid extraction and analysis[25–27].

With these issues in mind, we developed the software phactor™ to streamline the collection of HTE reaction data. Our primary objective was to develop a robust yet generalizable HTE workflow solution that captures the nuances of chemical experimentation while reporting

[1]Department of Medicinal Chemistry, University of Michigan, Ann Arbor, MI, USA. [2]Department of Chemistry, University of Michigan, Ann Arbor, MI, USA. [3]Natural Products Discovery Core, Life Sciences Institute, University of Michigan, Ann Arbor, MI, USA. ✉e-mail: tcernak@umich.edu

data in a standardized, machine-readable format. phactor™ minimizes the time and resources spent between experiment ideation and result interpretation. This enables creativity by freeing up time otherwise used thinking about experiment logistics, facilitates reaction discovery and optimization, and serves as a tool to bolster the amount of available reaction data reported in a standardized format. We have provided phactor™ as a free web service to the academic community, currently supporting 24- and 96-well formats, which can be accessed at https://phactor.cernaklab.com.

## Results

### phactor™ workflow overview

The workflow of a typical high-throughput experiment involves the design of the reaction array, preparation of reagent stock solutions, dosing of stock solutions according to the reaction array recipe (either by hand or with robotics), and evaluation of reaction outcome, followed by visualization and analysis of data and documentation of results. A standardized reaction template classifies substrates, reagents, and products (Fig. 1a). Interconnecting experimental results with online chemical inventories through this shared data format creates a closed-loop workflow for HTE-driven chemical research (Fig. 1b) and enables rapid reaction array design and analytics. While developing phactor™, we sought to maximize the automation of data movement and processing. Recognizing the rapidly accelerating chemical research software ecosystem[28–33,36–50], the philosophy behind phactor™'s data structure was to record experimental procedures and results in a machine-readable yet simple, robust, and abstractable format to naturally translate to other system languages (Supplementary Fig. 8). As such, the inputs and outputs of phactor™ can be procedurally generated or modified with basic Excel or Python knowledge to interface with any robot, analytical instrument, software, or custom chemical inventory containing metadata (e.g., the organophosphorous ligand platform Kraken[51]) such as reagent location, molecular weight, CAS number or SMILES string. Examples of interfacing phactor™ outputs with ORD[20], XDL[50], or EDBO+[48] are shown in Supplementary Figs. 14–16.

The event workflow of a typical phactor™ experiment is shown in Fig. 1c. The user selects desired reagents from the inventory for automatic field population or enters specific reagent entries manually, such as for a custom substrate. Once all relevant chemicals are selected, the reaction array layout is designed automatically or manually, as the user prefers. Reagent distribution instructions are generated to be executed either manually or by an interfacing liquid handling robot. Last-minute changes in the face of unforeseen circumstances during reaction setups, such as poor chemical solubility, chemical instability, or the need to premix reagents before dosing can be made at any time. Stock solutions are prepared in vials or wellplates and distributed to their respective locations on the reaction wellplate. Once the reactions are complete, they are quenched and analysed. Any data with a well-location map can be uploaded. This allows both data on reaction performance (e.g., UPLC-MS conversion) and biological assay results (e.g., bioactivity data) to be viewed in concert.

The workflows for executing HTE can vary depending on available equipment and desired throughput of the experiment. phactor™ incorporates these parameters into its user interface to ensure a consistent workflow experience agnostic of hardware capabilities (Fig. 1d). Examples herein demonstrate phactor™'s use with manual dosing as well as integration with the Opentrons OT-2 liquid handling robot for experiments of 384-well throughput or less, and the SPT Labtech mosquito® robot for 1536-well ultraHTE (Fig. 1e). Regardless of instrumentation or throughput, all results are stored in the same format, facilitating analysis of results across multiple experiments. Reaction discovery and library synthesis campaigns utilizing standard 24-, 96-, 384- and 1536-well experiments are described in detail[52–57].

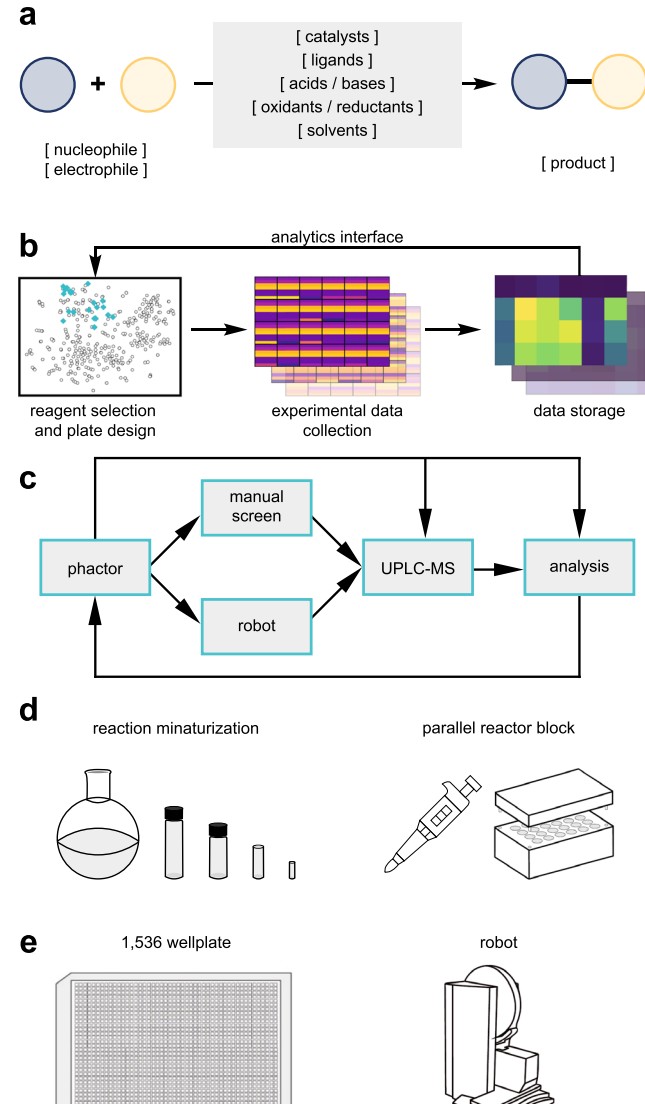

**Fig. 1 | Overview of the phactor™ software. a** Anatomy of a reaction as encoded by phactor™. **b** High-level software workflow of phactor™. Reaction arrays are designed from chemical inventories and reaction informatics. Resultant data is stored in delimited text (CSV) or in a relational database (SQLite3). phactor™ can convert results to Open Reaction Database (ORD)[20] and Chemical Description Language (XDL)[50] and is readily compatible with optimization programs such as EDBO+[48] and LabMate.ML[49]. **c** Workflow of phactor™. Once the reaction array is designed, phactor™ provides human-readable or machine instructions to execute the dosing manually or robotically (UPLC ultra-performance liquid chromatography). **d** phactor™ supports custom volumes allowing for reaction arrays to be performed at any scale. At a minimum, the hardware needed to execute a reaction array is an autopipette and an array reactor block. **e** phactor™ facilitates the design and execution of ultraHTE in 1536 wellplates.

## Experimental analysis

Phactor™ facilitates the discovery of new reactivity. Our lab is broadly interested in amine-acid coupling reactions[52,53,55–57] and particularly amine-acid C–C coupling reactions[55,57]. Diverse chemistries discovered with the aid of phactor™ are shown in Fig. 2. Figure 2a shows the discovery of a deaminative aryl esterification[53]. In the reaction array design, an amine, activated as its diazonium salt (**1**), a carboxylic acid (**2**), one of three transition metal catalysts, with one of four ligands, in the presence or absence of a silver nitrate additive was to be dosed to each reaction well in acetonitrile, then stirred at 60 °C for 18 h. phactor™ automatically designed the reagent distribution recipe by

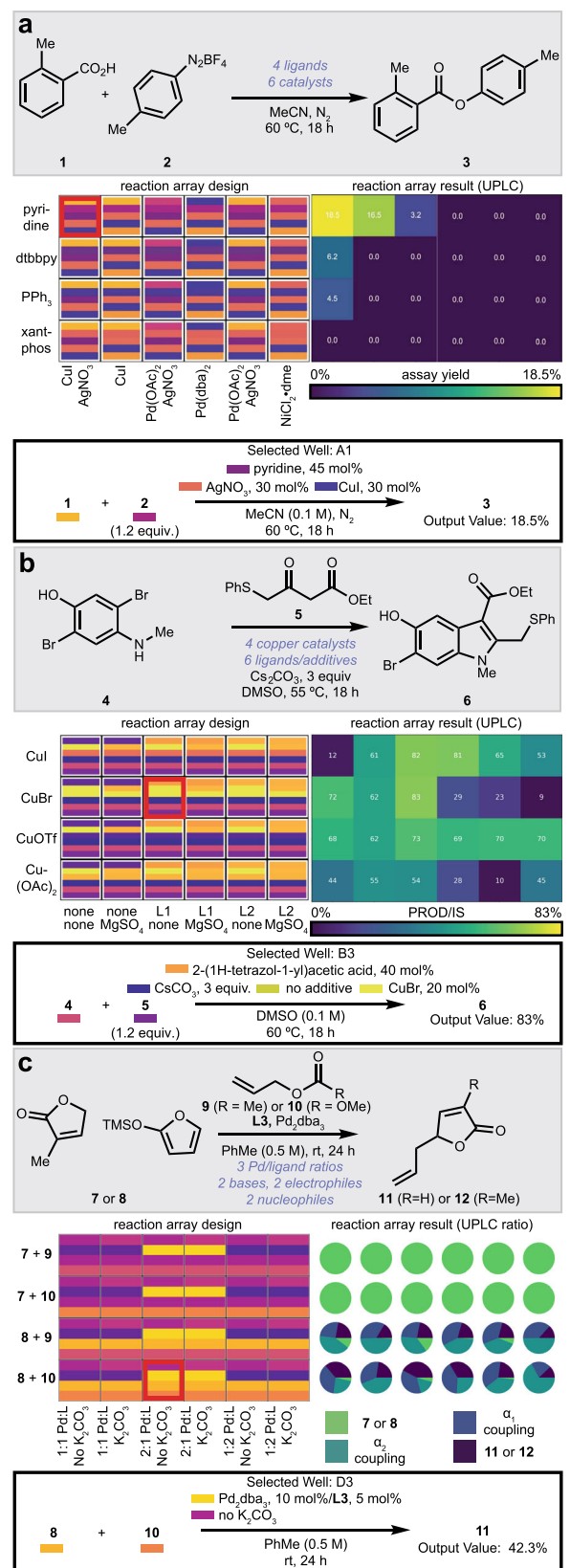

**Fig. 2 | Reaction arrays executed with the phactor™ software.** The reaction array design and results are shown here as displayed on phactor™. Colour bars adjacent to compound numbers correspond to the colour bars in the reaction array design grid generated by phactor™. Product/internal standard ratios are calculated using the observed UV-derived peak area, while assay yields account for differences in product absorptivity by calibrating to authentic samples of products. **a** Preliminary esterification hit leading to publication[53]. **b** Optimized oxidative indolization conditions towards the synthesis of umifenovir[54]. **c** Allylation catalyst/ligand concentration ratio and base reaction array analysed by conversion and selectivity.

UPLC–MS output files were analysed by the commercial software Virscidian Analytical Studio, which provided a CSV file containing peak integration values for each of the 24 chromatographic traces. This file was fed into phactor™ to record the experimental outcome and produce the heatmap shown in Fig. 2a. Analysis on phactor™ indicated an 18.5% assay yield when using 30 mol% CuI, pyridine, and AgNO₃, and these specific conditions were triaged for further study.

In the example of Fig. 2b, we optimized the penultimate step in our synthesis of umifenovir[54], an oxidative indolization reaction between **4** and **5** to produce **6**. Inspired by the conditions of Glorius[58], a reaction array was performed using copper catalysts and ligand/additive combinations. Four copper sources at 20 mol%: cuprous iodide, cuprous bromide, tetrakis(acetonitrile) copper(I) triflate, or cupric acetate, were distributed into the four rows while combinations of magnesium sulfate (0.0 equiv or 1.0 equiv) with 2-(1H-tetrazol-1-yl) acetic acid (**L1**), or 2,6-dimethylanilino(oxo)acetic acid (**L2**) at 40 mol% were distributed into the columns as DMSO solutions, with 3.0 equivalents of caesium carbonate added to every well as a suspension in DMSO. The reactions were manually arrayed in a glovebox, sealed, and stirred at 55 °C for 18 h. Well B3 (copper bromide with **L1** and no magnesium sulfate) was found to perform best, and a 0.10 mmol scale-up reaction produced desired indole **6** in 66% isolated yield.

In Fig. 2c, the allylation of furanone **7** or furan **8** with reagents **9** or **10** was investigated. For each combination of nucleophile and electrophile, one of three ratios of Pd₂dba₃ to (*S,S*)-DACH-phenyl Trost ligand **L3** was added, followed by the addition or omission of potassium carbonate as a base. Each reaction was run in toluene for 24 h at room temperature, quenched, and then analysed by UPLC–MS for conversion and selectivity. Multiplexed pie charts generated by phactor™ revealed that the conditions of well D3, with a 2:1 palladium catalyst to ligand loading and no base, generated the desired γ-regioisomer with the greatest selectivity, along with α-allylation and its olefin isomer when **8** was used.

An organocatalyzed asymmetric Mannich reaction is detailed in Supplementary Fig. 17. Aldehyde **13**, *p*-anisidine (**14**), and ketone **15** are used to form desired product **16** via a solvent and catalyst reaction array that was analysed by TLC, which revealed the formation of undesired product **17**.

With phactor™, HTE becomes an exercise in workflow execution, with automation of the organizational aspect of the experiment. This allows chemists to focus on the design and analysis of the reaction array, rather than workflow details (Fig. 3). Figure 3a–c displays three examples of 24-well experiments. Figure 3a shows an amide coupling reaction array performed in preparation for an ultra-high throughput direct-to-biology assay (vide infra), with the aim of producing inhibitors of the SARS-CoV-2 main protease (Mᴾʳᵒ)[59]. A screen of three anilines, two coupling agents HATU or DCC/HOBt, and a carboxylic acid, with or without base, produced all desired products. Amide **18** was produced in high conversion using HATU, DMAP, and DIPEA. Figure 3b and c detail the optimization of a recently published deaminative decarboxylative *sp²–sp³* C–C coupling from an acid-activated as an *N*-acyl-glutarimide and an amine activated as a Katritzky salt[55]. Reaction array 3b screened 24 ligands in NMP under nickel catalysis to generate product **19**, which was subsequently found with improved yield when using a bipyridine ligand and phthalimide as additives (Fig. 3c).

splitting the plate into a simple four-row and six-column multiplexed array. At completion, a solution containing one molar equivalent of caffeine was added to each well as an internal standard. An aliquot of each reaction was transferred to a plastic wellplate, then diluted with acetonitrile for UPLC–MS analysis of the desired ester product (**3**).

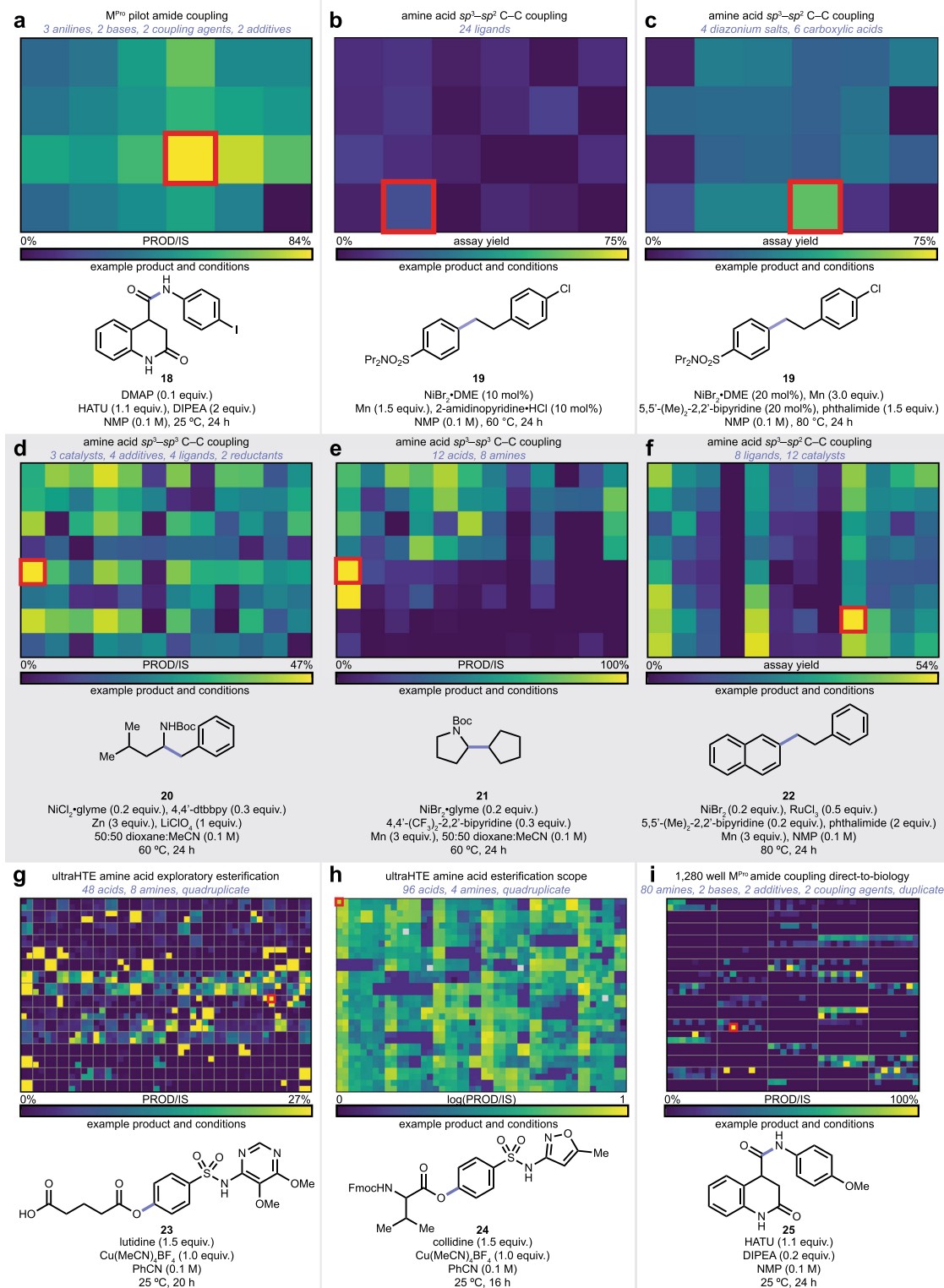

**Fig. 3 | phactor™ has been utilized in a variety of synthetic campaigns.**
**a**–**i** Chemistries discovered via reaction arrays designed with phac-
tor™. All input and output files used to produce reaction arrays (**a**–**f**) are provided via an online repository in addition to all compiled HTE results in a machine-readable format. Reaction schemes can be found in the "Selected screening examples" section of the Supplementary Information.

Figure 3d–f contains 96-well experiments designed with phac-tor™. Figure 3d and e show the results from a recently reported deaminative-decarboxylative $sp^3$–$sp^3$ C–C coupling[55] that generated products **20** and **21**. Three catalysts, four additives, four ligands, and two reductants were tested in Fig. 3d, and it was found that **20** was afforded at 47% product/internal standard with NiCl₂•glyme as the

catalyst, 4,4'-di-*tert*-butyl-2,2-bipyridine as ligand, zinc as the reduc-tant, and lithium perchlorate as an additive. After additional optimi-zation, the reaction array in Fig. 3e was run to test the reaction's scope. Product **21** was afforded with optimal conditions of NiBr₂•glyme, 4,4'-bis-trifluoromethyl-2,2'-bipyridine, and manganese in 1:1 dioxane:ace-tonitrile. Figure 3f shows the results of a 12-ligand−eight metal catalyst

reaction array further developing the analogous $sp^2$–$sp^3$ decarboxylative deaminative C–C coupling, optimizing the yield of model product **22**[57].

Figure 3g–i show several ultraHTE reaction arrays. Figure 3g and h are reaction arrays produced in the development of our aryl amine esterification reaction[53]. Both reaction arrays were substrate-scope experiments, producing ester compounds such as **23** and **24**. In Fig. 3g, lutidine and Cu(MeCN)$_4$BF$_4$ in benzonitrile showed good scope. Replacing lutidine with collidine expanded the reaction scope and reproducibility across quadruplicate measurements as shown in Fig. 3h. Finally, Fig. 3i contains the UPLC–MS results of a direct-to-biology reaction array towards the creation of SARS-CoV-2 M$^{Pro}$ inhibitors, generating amides such as **25**. Files needed for all 24- and 96-well experiments shown in Fig. 3 are provided as examples to learn the phactor™ workflow. The layouts for several reaction arrays are displayed in the Provided Examples section of the Supplementary Information.

**Discovery of a low micromolar inhibitor of SARS-CoV-2 M$^{Pro}$ via direct-to-biology assay.** An amide coupling experiment was planned based on an inventory of amines, largely anilines, and a tetrahydroquinoline carboxylic acid pharmacophore (**26**) found to be a potent inhibitor of M$^{Pro}$[59]. A preliminary 24-well amide coupling reaction array was performed to test the effectiveness of various amide coupling conditions for the acid and anilines (see Fig. 3a and Supplementary Fig. 21). A diluted aliquot from each well was subjected to a RapidFire™ MS binding assay to determine concentration-response curves[60]. Curves were found to correlate with yield and literature reported IC$_{50}$ values were replicated utilizing this assay[59]. Following an optimized direct-to-biology ultraHTE workflow (Fig. 4a), amide coupling reactions were executed with the aim of making diverse amides, which were directly tested for activity against M$^{Pro}$ in a single experiment. Conducting an experiment of this complexity using spreadsheet software would be challenging. However, with the use of phactor™, ultraHTE, and direct-to-biology experiments can be swiftly developed and assessed within a matter of minutes, especially if corresponding input or inventory files are in hand. Eight amide coupling conditions were tested in duplicate for each of the 80 amines, resulting in 1280 reactions (Fig. 4b). A key step in this workflow is the distribution of the 1536-well reaction plate into four 384-well analysis assay plates suitable for UPLC–MS or RapidFire™ analysis. As such, some of the wells in the 1536-well reaction plate were not utilized to account for the four control columns necessary in each of the four 384-well RapidFire™ assay plates to allow the calculation of Z prime for the assay (0.961)[61]. This distribution as well as the chemical and biological assay results are shown in Fig. 4c and d. Additional data analyses comparing chemical yield to biological response are shown in Supplementary Figs. 24 and 25. We note that reproducibility is a common concern in HTE and ultraHTE, and analyses of repeat experiments are provided in Supplementary Fig. 26. Both chemistry and biology assays are shown to be consistent, with 87% and 93% of data points having <10% error in the respective assays. From these analyses, three amides (**27**–**29**) were chosen for scale-up and IC$_{50}$ determination, two of which (**27** and **28**) were previously unreported in the literature. Compound **28** was found to have an IC$_{50}$ of 5.06 μM (Fig. 4e), competitive with the best-known M$^{Pro}$ inhibitors in this series[59]. Notably, IC$_{50}$ trends from pure compounds isolated on a larger reaction scale are well matched to the percent inhibition trends obtained on the nanomole scale.

## Methods

### General phactor™ workflow for general use and to recreate the chemistry reported in this and other manuscripts[52–57]

The workflow for phactor™ contains six stages: Settings, Factors, Chemicals, Analysis, and Report (Fig. 5). The first stage, *Settings*, simply begins the plate development process when provided an experiment name, reaction volume, and throughput (Supplementary Fig. 1a). The remaining stages can be optionally expedited with pre-generated workflow files that match a specified format.

The workflow files needed to reproduce the chemistry shown in Figs. 2a, b, and 3a–f are provided in an online repository. To replicate these studies, a 24- or 96-well reactor block, the corresponding amount of glass vials and stirbars, a hotplate with magnetic stirring, and autopipette will be required. The necessary solvents and reagents will also be required to the amounts as calculated on phactor™. Detailed specifications of hardware and chemical sources are provided in the "Experimental" section of the Supplementary Information. Having these files in hand provides an expedited utilization of phactor™—the user can directly drop any file that matches the chemical input specification directly into the Chemicals stage and click 'Run' to instantly get the procedural instructions for the experiments. Likewise dropping the provided corresponding analytical file into Analysis will display the reported results of the experiment.

Phactor™ can automatically distribute a list of chemicals into a multiplexed reaction array given that the chemicals are labelled as one of the reagent classes within the set {Electrophile, Nucleophile, Catalyst1, Catalyst2, Ligand1, Ligand2, Base/Acid, Reductant/Oxidant, Solvent1, Additive, or Other} and that the product of the amounts of each reagent class within the list is equal to or less than the wellplate size. For instance, a list containing six electrophiles, four nucleophiles, and a solvent can be automatically "plated" by phactor™. The desired plate design is specified in the *Factors* stage (Supplementary Fig. 1b).

Once the factors have been input and saved, the user can add reagents to be used in the experiment in the *Chemicals* stage (Supplementary Fig. 1c). These reagents can be either input by hand via textbox, where the user personally reports the reagent name, molecular weight, reagent class label, and density or reagents can be added directly from an online inventory. All chemicals are associated with a colour, which can be changed by clicking on the chemical's corresponding colour bar in the table. If factors were specified in the previous stage, a checklist is provided to ensure the requirements for the automatic distribution are met. Once all lines are green, hitting run will design the plate on the following Grid stage. A third alternative to populating reagents is to upload a CSV file containing reagent data in a specified format. As mentioned, example files corresponding to the experiments in this manuscript are provided in an online repository. In the case where the input reagent file contains a distribution of reagents that would perfectly fill out the reaction plate, the factors are automatically updated and thus factors need not be specified in the factors stage.

Following the *Chemicals* stage is the *Grid* stage, which is prepopulated if the automatic distribution feature was utilized. At this stage, an interactive grid allows the user to add or remove reagents to wells in bulk. Individual wells in the digital reaction array can be selected to view reaction details. In addition to the reaction array grid, a table containing the recipes for stock solutions, which indicate the mass of reagent and volume of solvent to be added to each source vial. The volume and locations of aliquots from each source vial are shown as well, to be executed via autopipette or liquid handling robot. Finally, a suite of buttons interfacing the designed reaction array with a variety of hardware and software is provided. On the provided version, MassLynx and Virscidian files are preformatted and provided with pre-populated metadata to facilitate the characterization of the reaction array.

When the reaction array is fully designed, the recipe can be downloaded. Stock solutions are dosed according to the generated procedure, and the reaction plate is moved to a reactor with the appropriate atmosphere, pressure, and temperature. After the designated reaction time has elapsed, the reactions are quenched, and aliquots of each reaction are transferred to an analytical plate containing internal standard in a UPLC–MS compatible solvent. Once the

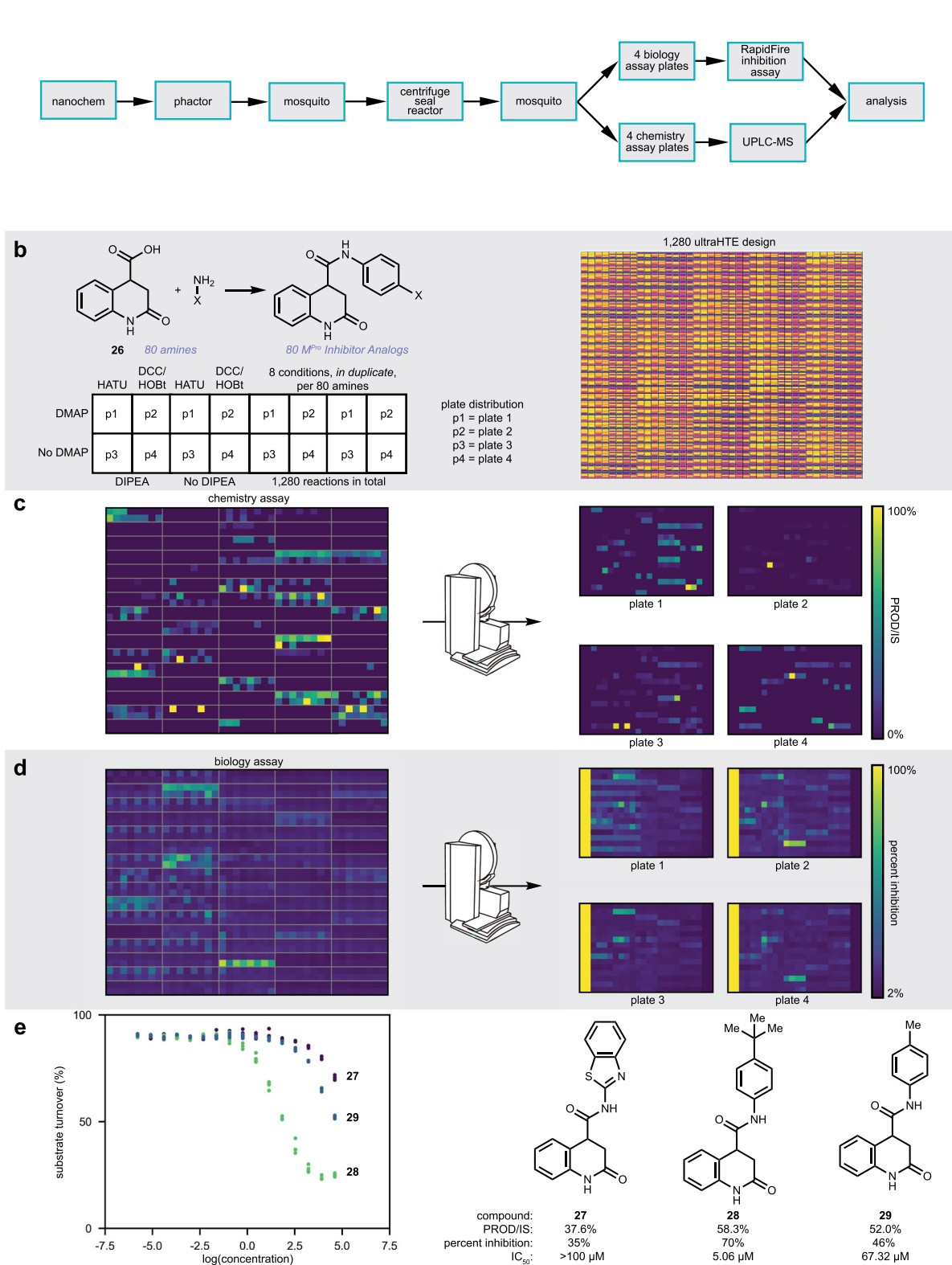

**Fig. 4 | phactor™ facilitates ultraHTE direct-to-biology campaigns. a** Event workflow for performing ultraHTE using phactor™ and a Mosquito robot. **b** Design of 1280 well amide coupling plate. 80 amines were selected to react with carboxylic acid **26**. Eight conditions were run in duplicate for each amine. **c** Results of the amide coupling are shown as a product/internal standard integration ratio from a 2-min LCMS injection of each well. The Mosquito robot is utilized to split the size 1536 plate into four sizes 384 plates for LCMS and bioassay analysis. **d** Percent inhibition of SARS-CoV-2 M^Pro when treated with a sample of the reaction mixture from the corresponding well. The 1280 plate is visually recreated. **e** IC$_{50}$ curves for three scaled-up compounds chosen from the reaction array. Compounds **27**–**29** display a range of assay and inhibitory responses.

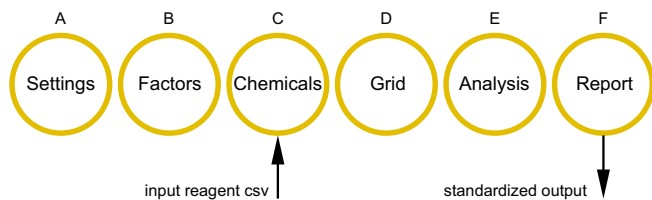

**Fig. 5 | The six stages in the phactor™ workflow.** Each stage is progressed sequentially. With an input reagent CSV, reaction arrays can be designed in seconds. Once the experiment has been executed, a standardized output can be downloaded on the report stage.

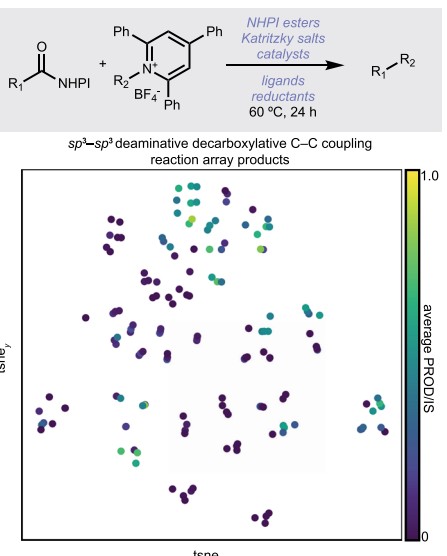

**Fig. 6 | phactor™ enables rapid machine-learning analysis of multiple reaction arrays in tandem.** Standardized output files can be rapidly merged to create massive datasets. Shown is a tSNE (t-distributed stochastic neighbour embedding) of all products made in the decarboxylative–deaminative $sp^3$–$sp^3$ C–C coupling detailed in ref. 55, coloured by average product/internal standard.

analytical plate has been characterized, the resultant file containing well locations and output results can be dropped into the *Analysis* stage of phactor™. Here an interactive triptych display allows the user to view several output results in the same view. Clicking on a well displays the input and outputs of the reaction, as well as an image of the product if specified.

The final stage is *Report*. An overall one-page summary of the results is shown here via visualizations of inputs and outputs. Additionally, statistics of the results are calculated, and the top-performing reagents are shown. Most importantly, here is where the user can download the compiled results of the entirety of the reaction array in a single CSV file. This file contains the inputs and metadata associated with each reaction of the reaction array, including reagent labels, SMILES, molarities, and solvents, as well as the output values associated with each reaction. This file can be trivially collated with other plate outputs to make supermassive reaction datasets.

**Machine learning Python scripts for phactor™ output**

In the provided repository an example Python notebook is provided to facilitate downstream machine learning analyses from the phactor™ output file. In this example notebook, tSNE plots are generated from collated from a deaminative-decarboxylative $sp^3$–$sp^3$ C–C coupling optimization campaign dataset (Fig. 6). Product SMILES are encoded

into fingerprints using RDKit, and fingerprint matrices are fed into the tSNE algorithm to reduce the dimensionality of the dataset into two dimensions. These are subsequently plotted and coloured by the average output value of each product to provide a rapid visualization of the reaction manifold. Areas of reaction success and failure are then readily identified in a singular display. Notebooks for conversion to heatmaps and kernel density estimate plots are also provided. Sample notebooks converting phactor™ output to ORD, EDBO+, and XDL are also provided within the repository. All the provided scripts can be easily modified by changing the input file string to work with other datasets.

In conclusion, we present the HTE ELN phactor™, which records all details of an experiment to allow for robust reproduction and accelerated discovery. phactor™ stores all experimental details in a machine-readable yet tractable and interpretable format with an SQL database and on the cloud to facilitate the use of downstream statistical analysis. As all reaction arrays are stored in a centralized database, bulk analyses of multiple reaction arrays can be performed. phactor™ provides an exposed API that can be used to develop interfaces to other robots, assays, and software. Examples of various integrations and code infrastructures are shown in the "Discussion" section of the Supplementary Information. Furthermore, we are currently developing additional integrations with AI and ML models to generate and evaluate reaction arrays[62] and to discover additional amine acid couplings via automated mining of relevant literature[63–67]. We hope that phactor™'s ease of use provides increased accessibility to HTE and HTE data in the chemistry community. Registration-free and non-commercial use of phactor™ in 24- and 96-well formats is available through https://phactor.cernaklab.com/.

**Inclusion and ethics**

Local researchers were included throughout the entirety of the research process, and all roles and responsibilities were agreed to ahead of the research.

## Data availability

All sample files and data generated in this study have been deposited in an online GitHub repository[68]. https://github.com/cernaklab/public-phactor-example-files.

## Code availability

All relevant interfacing scripts generated in this study have been deposited in an online GitHub repository[68]. https://github.com/cernaklab/public-phactor-example-files.

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

## Acknowledgements

O.G.M. and A.T. thank Michigan Drug Discovery for supporting the M^Pro inhibition screening assay (MDD-22105S to A.T.). We would like to thank the Center for Structural Biology and the Proteomics and Peptide Synthesis core at the University of Michigan for supplying the M^Pro enzyme and the substrate peptide for the M^Pro inhibition assay, respectively. We would like to thank the Center for Chemical Genomics at the University of Michigan for providing access to different instruments. We gratefully acknowledge funding from MilliporeSigma (Y.L.), Relay Therapeutics (J.D.), Janssen Therapeutics (A.M.), the National Science Foundation (CHE-2236215) (T.C.) an ACS MEDI Predoctoral Fellowship (B.M.), as well as generous gifts of liquid handling automation from SPT Labtech and Merck & Co., Inc. Initial studies were funded by startup funds from the University of Michigan College of Pharmacy. We gratefully acknowledge funding from Schmidt Futures.

## Author contributions

B.M. wrote and deployed the software and developed, performed, and analysed chemical and biological experiments. B.M., Y.S., Y.L., and R.Zhao. performed chemical experiments. O.G.M. and A.T. performed the inhibition assay. B.M. and T.C. wrote the manuscript with input from all authors. T.C. conceived and supervised the project. R.Z, Y.S, A.M., Y.L., Z.Z, J.L.D., and T.C. provided feedback on software development. All authors have given approval to the final version of the manuscript.

## Competing interests

The Cernak Lab has received research funding or in-kind donations from MilliporeSigma, Relay Therapeutics, Janssen Therapeutics, SPT Labtech, and Merck & Co., Inc. T.C. holds equity in Scorpion Therapeutics and is a cofounder of and equity holder in Entos, Inc. The remaining authors declare no competing interests.
