## [Peer Review File · Nature Communications]

REVIEWER COMMENTS

Reviewer #1 (Remarks to the Author):

This manuscript describes a software for planning and analyzing high-throughput experimentation (HTE) reactions, as well as several specific applications of that software.

HTE is an important technology for accelerated reaction discovery/optimization and reproducible data generation and as such a key component of modern chemistry. Planning and analyzing HTE reactions can be very tedious processes that may well take much longer than setting up the screening reactions themselves. As such, the software described in this article addresses an important need for a highly relevant field and I fully recommend a publication of this manuscript in Nature Communications.

The manuscript conclusively shows the utility of the phactor software with a range of application case studies. If anything, I would say that the discussion of some of the case studies is unnecessarily detailed, considering that the focus of the manuscript is the software and not HTE screening itself. For example, complete reaction procedures with exact concentrations, internal standard additions etc are explained several times.

I thought that the ultraHTE direct-to-biology assay is the most impressive demonstration, in part due to the distribution of experiments over two separate screening assays with different-sized plates. This might be a place to comment on a few practicalities such as the typical time spent in phactor to plan/execute this, and maybe how long that would take with other tools?

I appreciate the availability of scripts for followup tasks such as machine learning or conversion to ORD format. Considering the importance of the ORD vision for the modeling community, perhaps this article could contain a statement to actively encourage users of phactor to submit their results to the ORD?

Minor notes:

- In Fig 2 A,B what is the meaning of the colors? And in C why are there two sections for each substrate combination? Please expand the figure caption accordingly.

- There are some typos here and there. A few that I took note of:

page 1 left "detailed reaction data must be easily accessible for and standardized rapid extraction"
typo first "and"

page 3 left "At competition" - completion, I take it?

page 7 left "amide reactions were executed with the aim of making diverse amides in which were directly tested for activity against MPro in a single experiment." probably an excess "in", please check and revise

- The discussion on page 7 talks about "amide reaction" and "amide conditions" repeatedly. Should be amide coupling reaction, amide coupling conditions.

Reviewer #2 (Remarks to the Author):

Comments for the authors,

This manuscript deals with the fundamental build of a data management system/workflow for HTE. And as such its is highly useful for an HTE focused audience. It stands on its own really, and the interesting part is the workflow but also the benefits Phactor brings in comparison to more elaborate ways of evaluating and visualizing HTE experiments. Phactor is available free of charge for academics and non-commercial use - great for the field of HTE!.

What are the noteworthy results, aim and goal for this manuscript?

This is atypical in its format and as such harder to judge from a scientific standpoint. The Introduction reads well, and appropriate references are present. The following section "RESULTS AND DISCUSSION" appears more like an "accounts" of the very nice activities largely already published by the Cernak group. However the it seems like Phactor facilitates and catalyzes the visualization really well. What I like the most and that really merits publication is the methods section - along with the much needed tutorial section to be found in the SI.

Furthermore there is a section dealing with the link to making data machine readable for AI/ML proposes - again this is great to see! And I don't know of anyone who clearly build ready to use tools for this in HTE/chemistry. Are there any examples of ML learning exercises from any of the HTE experiments? If so, that would add greatly to the manuscript. Finally the conclusion is adequate but does not add much value - maybe would benefit from comparisons to other ways of analyzing HTE data? but also from describing future development efforts. I would consider redrafting it to capture the key advancements Phactor brings.

Will the work be of significance to the field and related fields?

It is certainly of value for a specialized HTE lab, and scientists who want to develop into this area.

Is there enough detail provided in the methods for the work to be reproduced?

I believe so, having that said I have not downloaded Phactor to try it out. The SI is clearly written and the tutorial part is well outlined.

Reviewer #3 (Remarks to the Author):

This manuscript describes the application of HTE ELN phactor™ a new software in-house developed by the Cernak's labs for designing, preparing, recording, and analyzing (ultra-) HTE experiments. Very interestingly, the software seems to be quite flexible and provides easy interfacing to robotic platforms, assays, and others software and can be of great interest for the HTE community. The fact that the output of reaction data is in a standardized format is of special value in the light of more easy sharing of datasets for ML purposes.

I would recommend to publish it after the following comments are resolved

1) In the introduction, I believe it would be important and fair to comment on existing solutions for HTE such as Katalyst D2D from ACD/Labs or LEA from Unchainedlabs

2) phactor™ seems to be a free web service to the academic community only for the 24/96 format. Please explain on which basis this choice was made. Much more benefit could be gained for higher format designs (384+) where the interfacing with pipettors and data mining is more important and challenging.

3) I understand that the "reaction array design" of (for example) Fig 2C is an automated output of Phactor. Still, these pictures are quite complex to be fully understood. In other words: what are the colors exactly referring to? For example, in the specific example of Fig 2C there are 8 rows but only 4 isoquinolines. I guess a dimension is missing e.g. solvent ?.

To me, this risks to be a general issue for all the presented "reaction array designs"

My suggestion would be to remove all of them or give more insights to facilitate the interpretability.

In this light, it would be beneficial also to be able to control in Phactor the color assigned to each component in the graphical interface to improve the readability of the reaction arrays

4) "At competition, a solution containing..." should be "at completion, a solution..."

5) "quadruplicate measurements as shown in Fig. 4F" should be "in Fig 4G-H"

6) "General phactor™ workflow for general use and to recreate the chemistry reported in this and other manuscripts"

Here appropriate references should be provided

7) "The workflow for phactor™ contains six stages: Settings, Factors, Chemicals, Analysis, and Report (Extended Data Fig. 5)"

In the SI stages are named differently (Create, Factors, Chemicals, Grid, Analysis, and Report) you may want to align for clarity

8) In the method part for this workflow a diagram would be of great help describing the different stages of Phactor. In particular, it would be good to have a picture reporting all the "Create, Factors, Chemicals, Grid, Analysis, and Report" stages and details of possible inputs (e.g. CSV file) and outputs for each stage

Response to the reviewers for the manuscript: "Rapid Planning and Analysis of High-Throughput Experiment Arrays for Reaction Discovery"

Babak Mahjour¹, Rui Zhang², Yuning Shen¹, Andrew McGrath¹, Ruheng Zhao¹, Osama G. Mohamed³, Yingfu Lin¹, Ziron Zhang¹, James L. Douthwaite¹, Ashootosh Tripathi^{1,3}, Tim Cernak^{*1,2}

¹Department of Medicinal Chemistry, University of Michigan

²Department of Chemistry, University of Michigan

³Natural Products Discovery Core, Life Sciences Institute, University of Michigan

Response to Referee 1:

We thank the reviewer for their kind remarks.

We agree that the chemistry is written with much detail and hope that the manuscript conveys the breadth and depth in which phactor can operate. We have removed some chemistries from the main text to improve its focus.

We now emphasize the speed in which phactor can manage ultraHTE by adding the following: "Conducting an experiment of this nature using spreadsheet software would be impractical or require a substantial amount of time for planning and analysis. However, with the use of phactor™, ultraHTE and direct-to-biology experiments can be swiftly developed and assessed within a matter of minutes, especially if corresponding input or inventory files are in hand."

We now "encourage the reader and users of the software to reposit their experiments to the Open Reaction Database" in the conclusion of the text.

Minor Notes:

- The color bars next to the compound numbers to match with the colors in the plate design generated by phactor. We have removed this chemistry from the text, but the same pattern applies in the new figure 2. We now elaborate in the figure caption that "color bars adjacent to compound numbers correspond to the color bars in the reaction array design grid generated by phactor™."
- We are grateful to the reviewer for their careful consideration of the grammar in the manuscript, and we have addressed all the identified errors.

Response to Referee 2:

We greatly appreciate the reviewer's response.

We agree that the manuscript is somewhat atypical. We have renamed the "results and discussion" section to "case studies" to better fit the style of the text.

While we have yet to publish any ML studies on HTE data from phactor, many examples of ML publications on HTE data from other sources are cited in the paper. We look forward to sharing our results with the community soon, and hope that the data repositied alongside this manuscript will enable others to do the same!

We have redrafted the conclusion to now include:

"Using data that is stored in an unstructured format, such as data found in classical notebooks or literature reports, can be challenging due to the need for manual compilation and reorganization. To address these issues, phactor™ stores all experimental details in a machine-readable yet tractable and interpretable format with an SQL database and/or on the cloud to facilitate the use of downstream algorithms."

As well as "Furthermore, we are currently developing additional integrations with AI and ML models to generate and evaluate reaction arrays."

Response to Referee 3:

We thank the reviewer for their comments and suggestions.

1. We now refer to contemporary HTE software such as Katalyst D2D and LEA in the introduction.
2. We respond to the reviewer that 24- and 96- well formats require the lowest barrier to entry. Notably, any number of experiments up to 96 reactions can be designed with these formats.

3. We have removed this example to improve the focus of the manuscript and have standardized the presentation of reaction arrays. We agree that having control over the color is important, and it is possible to change the color in phactor by clicking on the color in the Chemicals stage. We have now noted this in the tutorial and methods sections.
4. We thank the reviewer for pointing out this error. It has been corrected.
5. We have corrected the figure reference in the text.
6. We have added the appropriate citations to the identified text
7. The correct stages are named "Settings, Factors, Chemicals, Analysis, and Report". We have aligned this in the SI.
8. We have added a workflow diagram summarizing the stages of the software in the main text.

REVIEWERS' COMMENTS

Reviewer #3 (Remarks to the Author):

The points highlighted in my initial review have been addressed by the authors and the manuscript can in my opinion be published as it is.